# Developmental assembly of multi-component polymer systems through interconnected synthetic gene networks in vitro

Daniela Sorrentino [1,2], Simona Ranallo[2], Francesco Ricci [2] ✉ & Elisa Franco [1] ✉

Living cells regulate the dynamics of developmental events through interconnected signaling systems that activate and deactivate inert precursors. This suggests that similarly, synthetic biomaterials could be designed to develop over time by using chemical reaction networks to regulate the availability of assembling components. Here we demonstrate how the sequential activation or deactivation of distinct DNA building blocks can be modularly coordinated to form distinct populations of self-assembling polymers using a transcriptional signaling cascade of synthetic genes. Our building blocks are DNA tiles that polymerize into nanotubes, and whose assembly can be controlled by RNA molecules produced by synthetic genes that target the tile interaction domains. To achieve different RNA production rates, we use a strategy based on promoter "nicking" and strand displacement. By changing the way the genes are cascaded and the RNA levels, we demonstrate that we can obtain spatially and temporally different outcomes in nanotube assembly, including random DNA polymers, block polymers, and as well as distinct autonomous formation and dissolution of distinct polymer populations. Our work demonstrates a way to construct autonomous supramolecular materials whose properties depend on the timing of molecular instructions for self-assembly, and can be immediately extended to a variety of other nucleic acid circuits and assemblies.

A key feature of biomolecular materials is their ability to operate out of equilibrium and adapt to fluctuating environmental conditions[1,2]. A classical example is given by cytoskeletal networks[3], which are composed of filamentous assemblies that form and dissolve dynamically in response to endogenous and exogenous signals[4,5]. Filament assembly organization is orchestrated by complex cellular signaling and regulatory networks that have evolved to interact synergistically, and determine when and where filaments will form[6]. Learning from Nature,

artificial biomolecular materials with the capacity to respond to stimuli have been built by rationally coupling components that self-assemble, and components that generate regulatory signals controlling the properties of self-assembling parts. In this context, approaches based on the use of nucleic acids to build both self-assembling elements and regulatory elements have been very productive[7–9]. Due to their predictable non-covalent interactions that can be programmed through sequence design, DNA and RNA are ideal materials to build assemblies

[1]Department of Mechanical and Aerospace Engineering, University of California at Los Angeles, Los Angeles, CA, USA. [2]Department of Chemical Science and Technologies, University of Rome, Tor Vergata, Via della Ricerca Scientifica, Rome, Italy. ✉e-mail: francesco.ricci@uniroma2.it; efranco@seas.ucla.edu

whose structural precision and complexity is approaching that of natural molecular machines[10–12]. At the same time, the binding kinetics and equilibria of nucleic acid molecules can be prescribed by tuning the affinity of base-pairing domains[13,14], making it possible to build molecular systems operating like logic[15,16] or dynamic circuits[17]. Networks that take advantage of energy-dissipating enzymes or catalytic processes have demonstrated an extensive range of non-equilibrium dynamic behaviors when compared to enzyme-free circuits[14,18]. Naturally, nucleic acid assemblies and nucleic acid circuits can be seamlessly integrated: molecules produced by a nucleic acid circuit can be designed to hybridize with the domains of a nanostructure, thereby influencing its stability and capacity to assemble. This principle has fueled the development of DNA and RNA machines and supramolecular materials operating under the control of DNA and RNA networks[19,20]. As major progress is being made toward building multi-component dissipative networks[16,21], and multi-component structures[22,23], questions are emerging about how to design network-structure interactions that are scalable and robust, and can achieve autonomous temporal behaviors.

Here, we propose a method to coordinate the assembly of distinct DNA components using modular synthetic genes to generate a transcriptional signaling cascade[24]. To achieve this, we have developed a platform of synthetic genes and DNA building blocks that can be modularly interconnected without crosstalk. Each gene of the cascade produces an RNA output carrying specific instructions to control a particular self-assembling component. The RNA output production rate can be fine-tuned through careful design of the gene. Further, the time at which the RNA output is released depends on the gene position in the cascade. Our building blocks are DNA tiles that polymerize into nanotubes[25], and whose assembly behavior can be controlled by RNA molecules targeting the tile interaction domains[7,26]. By using the same functional components, genes and tiles, but changing how genes are cascaded as well as their RNA output production speed, we demonstrate that we can obtain nanotube assembly outcomes that differ in composition and time, including random DNA polymers, block polymers, and as well as distinct autonomous formation and dissolution of distinct polymer populations (Fig. 1).

We envision that our approach can be easily extended to a variety of other nucleic acid circuits and assemblies[18,19], because the conditions required to support RNA transcription and degradation are immediately compatible with a variety of DNA-based devices. Beyond nanotechnology, our results illustrate how to develop autonomous supramolecular materials whose properties depend on the timing of biochemically released assembly instructions, so that the same components can be routed toward a different fate depending on how regulatory signals are integrated[27].

## Results

### Activation and inhibition of DNA building blocks

We consider double crossover DNA tiles composed of five distinct strands (Fig. 2a)[25,28,29]. Tiles interact via 5-nucleotide (nt) sticky ends leading to spontaneous self-assembly of micron-long nanotubes[28,30] that can be visualized by labeling individual tiles with fluorophores[25]. Building on previous work, we modified the tiles to be activated or inhibited by sequence-specific RNA strands that can be produced by synthetic genes (Supplementary Figs. 1, 2, 3)[7,31,32]. In our approach, one of the tile sticky ends is modified to include a 7-nt toehold (Fig. 2a, black domain) that serves as a binding domain for a single-stranded synthetic RNA inhibitor (Fig. 2a, light blue domain) complementary to both the toehold and sticky-end sequence (12 nt). The inhibitor not only prevents tile self-assembly, but it can also disassemble formed nanotubes due to the presence of the toehold which enables invasion of one of the tile sticky-ends (invasion of a single sticky-end is sufficient as tile assembly is a cooperative process). Therefore, addition of inhibitor results in nanotube disassembly within minutes (Fig. 2a)[7,33]. To activate the tiles and restore their ability to self-assemble, we employ an RNA activator designed to displace the inhibitor strand (Fig. 2a). The DNA sequences forming a tile are arbitrarily designed through computer programs[34,35], so one can generate an expandable set of sequence-distinct tile populations that can be individually controlled by their specific RNA activators and inhibitors[7,26].

### Synthetic genes can be designed to precisely tune the kinetics of tile activation

Building on previous work[26,31], we designed synthetic genes producing RNA activators and inhibitors in situ that target a specific tile and control its active or inactive state with the desired kinetics. Each gene is a linear template that includes the T7 RNA polymerase (T7 RNAP) promoter site (Fig. 2b) and a downstream region encoding its RNA transcript sequence (RNA output)[36,37]. Detailed schematics describing the composition of each gene are in Supplementary Figs. 4, 5, and 6. Because the typical transcription temperature (37 °C) exceeds the melting temperature of the DNA nanotubes considered here, all the transcription experiments in this work were done at 30 °C.

The speed of tile activation or inhibition can be modulated by changing the speed of RNA transcription. We illustrate this idea by measuring the kinetics of activation of DNA tile type 1 as a function of inhibitor, gene and T7 RNA polymerase concentration (Fig. 2b) (RNA-based tile inhibition has been characterized in ref.[7]). The DNA tile is initially bound to its synthetic RNA inhibitor 1 and is activated by an RNA activator 1. To assess the kinetics of activation, we labeled the RNA inhibitor strand with a fluorophore (Cy3) and the corresponding tile strand with a quencher (BHQ1). When the tiles are inactive, the strands labeled with the fluorophore and the quencher are in close proximity, resulting in a low fluorescence signal; fluorescence increases when the labeled inhibitor is displaced by transcribed RNA activator, which activates the tiles as shown in Fig. 2b. First, we checked in a control experiment that the addition of the synthetic RNA activator strand 1 to the inactive tiles causes their activation, as shown in Supplementary Information (Supplementary Fig. 7). The kinetics of tile activation are captured by a simple ordinary differential equation model (Supplementary Note 2), using unfitted reaction rate parameters that are comparable with values found in the literature[7] (Supplementary Fig. 39). Next, given a fixed amount of tiles (250 nM) and RNA inhibitor (1 μM), we controlled the tile activation speed by varying the

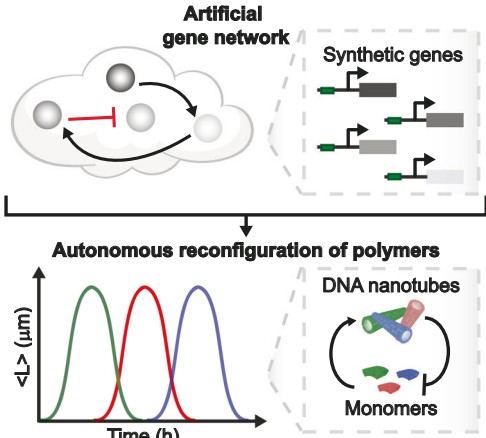

**Fig. 1 | We integrate artificial gene networks to regulate the assembly of distinct building blocks, routing the system to different assembly states over time.** We propose a scalable approach that integrates the principles of self-assembly and in vitro transcription to create a synthetic biopolymer system capable of programmed and autonomous reconfiguration. This is achieved through a suite of artificial genes and connectors that generate precise temporal instructions to activate or deactivate the assembly of distinct DNA-based monomers.

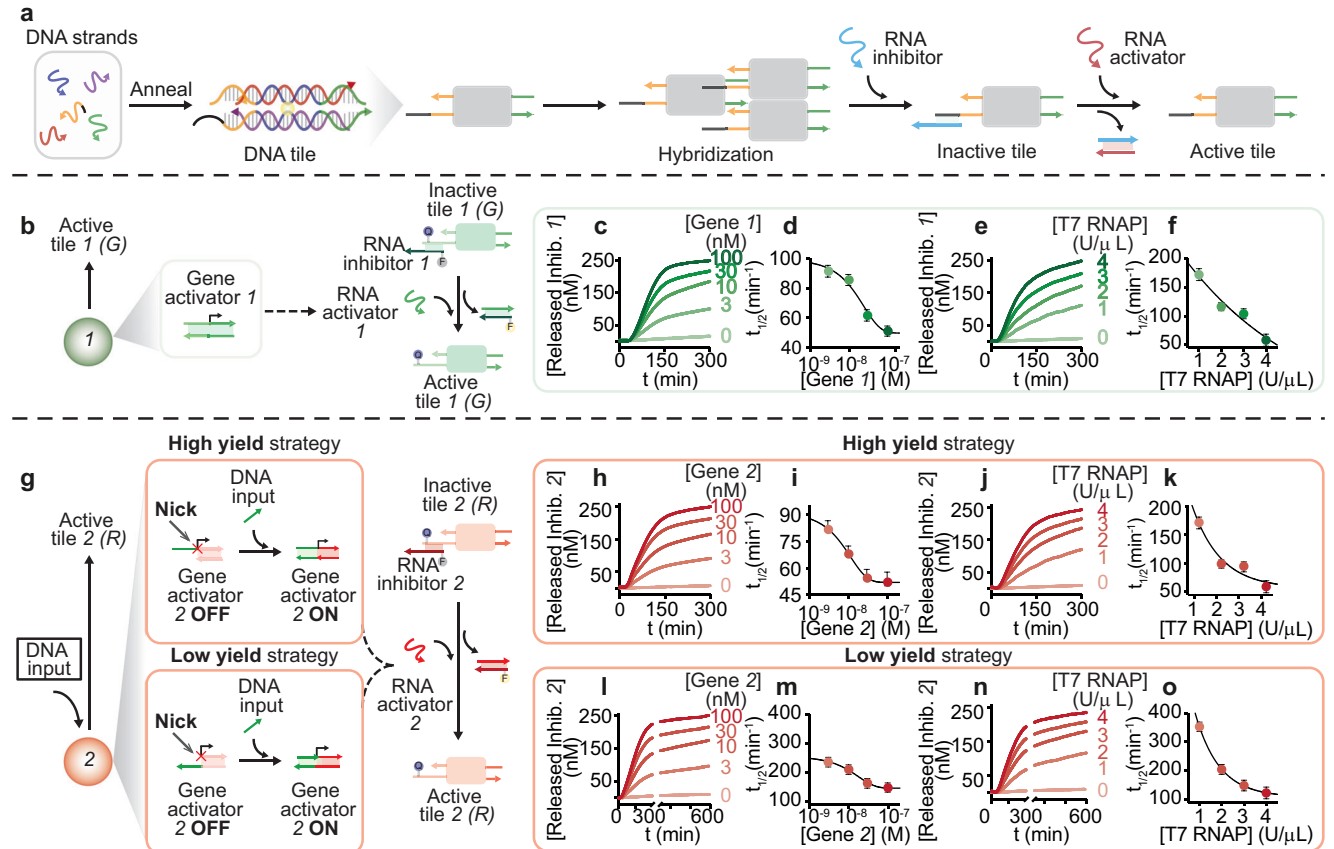

**Fig. 2 | Optimization of RNA-induced activation of different DNA tiles. a** DNA double crossover tile variants that self-assemble into nanotubes and contain a single-stranded overhang (toehold) portion to enable RNA inhibitor strand invasion and the following tile inactivation and nanotubes disassembly. The presence of an additional toehold domain into the RNA inhibitor strand allows tile reassembly. In all sketches, the 3′-ends are marked with an arrow. **b** Gene 1 (green filled circle) transcribes RNA activator strand 1, which activates tile 1. Fluorescence kinetics and reaction half-activation time ($t_{1/2}$) at different concentrations of gene 1 (**c**, **d**) and at different concentrations of T7 RNAP added to a solution containing a fixed amount of tile 1 (250 nM), RNA inhibitor strand 1 (1 µM), and gene 1 (100 nM) (**e**, **f**). **g** Gene 2 (red filled circle) transcribes RNA activator strand 2 with two different design strategies (high yield and low yield). Fluorescence kinetics and reaction half-activation time ($t_{1/2}$) at different concentrations of high yield synthetic gene 2 and equimolar amounts of DNA input to complete gene 2 (**h**, **i**) and at different concentrations of T7 RNAP added to a solution containing a fixed amount of tile 2

(250 nM), RNA inhibitor strand 2 (1 µM), gene 2 (100 nM), and DNA input (100 nM) (**j**, **k**). Fluorescence kinetics and reaction half-activation time ($t_{1/2}$) at different concentrations of low yield synthetic gene 2 and equimolar amounts of DNA input to complete gene 2 (**l**, **m**). **n** Fluorescence kinetics and **o** reaction half-activation time ($t_{1/2}$) obtained at different concentrations of T7 RNAP added to a solution containing a fixed amount of DNA tile 2 (250 nM), RNA inhibitor strand 2 (1 µM), gene 2 (100 nM), and DNA input (100 nM). Experiments shown in this and the following figures were performed in 1× TXN buffer (5× contains: 200 mM Tris–HCl, 30 mM MgCl₂, 50 mM DTT, 50 mM NaCl, and 10 mM spermidine), pH 8.0, 30 °C. Experimental values represent averages of three separate measurements; error bars represent standard deviation and the center of the error bars represents the mean. Black curves in (**d**, **e**, **m**) are logistic functions fitted to the data points as a guide to the eye; black curves in (**f**, **k**, **o**) are decaying exponential functions fitted to the data points (Supplementary Note 1).

concentration of gene producing RNA activator (from 0 to 100 nM), obtaining a clear concentration-dependent response (Fig. 2c), with half-activation time that decreases with gene activator amount ($t_{1/2} = 50 \pm 5$ min at 100 nM gene 1 concentration) (Fig. 2d). After 24 h, the inhibitor is fully released across all conditions (Supplementary Fig. 8). Similarly, given a fixed amount of activator gene (100 nM), we can vary the T7 RNA polymerase level (from 0 to 4 U/µL) to modulate the speed of tile activation, achieving half-activation times between ~50 and 180 min as shown in Fig. 2e, f. These experiments were also generally reproduced by a simple ODE model (Supplementary Figs.40 and 41), which suggests fast kinetic parameters for binding of RNAP and for tile activation.

Synthetic genes can be activated or deactivated through a promoter-displaced mechanism[38,39]. In this approach (Fig. 2g), a synthetic gene can take one of two different states, OFF (inactive) or ON (active). The gene state depends on whether the promoter is double stranded or partially single stranded: one of the strands of the promoter site is nicked at −12, and if the upstream region is removed the

gene is in an OFF state (negligible transcription). Transcription is switched ON if an ssDNA activator binds to and completes the promoter. When the gene is ON, we found that the transcription rate and yield depend on whether the nick is placed on the template strand (high yield) or on the non-template strand (low yield) (Supplementary Figs. 9–11). We take advantage of nick placement as a simple means to tune the kinetics of RNA transcription, and DNA tile activation, that does not require altering the level of gene, the level of enzyme, nor the promoter sequence.

In Fig. 2g, we demonstrate that a switchable synthetic gene can be used to obtain tunable tile activation kinetics; here we used a set of DNA tiles (red, 2) that have the same sticky ends as tile 1 (see above) but a different toehold binding domain that allows recognition of a different synthetic RNA inhibitor strand 2. Like done earlier, we labeled the RNA inhibitor 2 with a fluorophore (Cy5) and one of the tile strands with a quencher (BHQ2) to measure the activation kinetics (Fig. 2g). In a control experiment, we verified that addition of RNA activator 2 successfully activates the tiles (Supplementary Fig. 12). Next, we

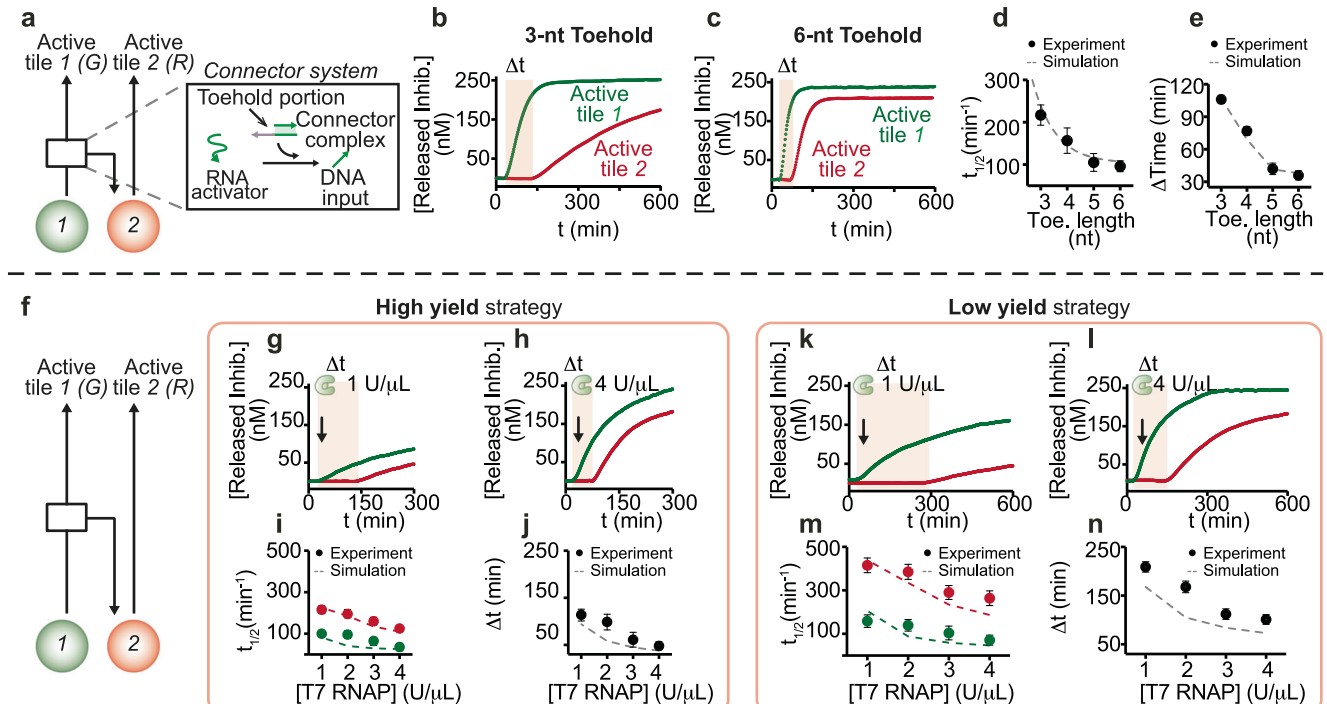

**Fig. 3 | Sequential control of assembly of distinct tiles through a cascade of synthetic genes. a** Schematic representation of how two different genes (1, green circle and 2, red circle) can be connected by a connector system. The second gene of the network is switched ON, when a DNA input is released from a connector complex. **b, c** Fluorescence kinetics of the release of RNA inhibitor strands 1 and 2 from their corresponding tiles by the presence of transcribed RNA activator strands 1 and 2, and connector complex at different toehold lengths (from 3 to 6 nt). **d** Reaction half-activation time ($t_{1/2}$) of inhibitor 2 released from DNA tiles 2 as a function of toehold length. **e** Delay of inhibitor 2 release as a function of toehold length. **f** The connector system allows sequential activation of two different tile types, each connected to a different gene (1, green, and 2, red), occurring at different times. **g, h** Fluorescence kinetics at different concentrations of T7 RNAP (from 0 to 4 U/μl), using a fixed concentration of high yield gene 2 (100 nM), connector complex (300 nM), and DNA input (300 nM). **i** Reaction half-activation

time ($t_{1/2}$) of inhibitors 1 (green) and 2 (red) released from DNA tiles as a function of T7 RNAP concentration. **j** The delay time at which tile 2 activation begins can be tuned by varying the concentration of T7 RNAP. **k, l** Fluorescence kinetics at different concentrations of T7 RNAP (from 0 to 4 U/μl) showing the displacement of RNA inhibitors 1 and 2 from the corresponding inactive tiles by the presence of the transcribed RNA activators, using a fixed concentration of low yield gene 2 (100 nM) and connector complex (300 nM), and DNA input (300 nM). **m** Released inhibitor from tiles 1 and 2 as a function of T7 RNAP concentration at 600 min reaction. **n** Time-dependent response of tile activation at different concentrations of T7 RNAP. In (**d, e, i, j, m, n**) dots represent experimental values, while dashed lines are obtained with an unfitted computational model (Supplementary Note 2). Experimental values represent averages of three separate measurements; error bars represent standard deviation and the center of the error bars represents the mean.

tested activation with switchable synthetic genes. When using the high yield gene 2 (nick on the template strand) we found that the speed of tile activation is comparable to the case in which we used constitutively active gene (Fig. 2b), both when we varied activator gene level and T7 RNAP level (DNA tiles 2 fixed at 250 nM and RNA inhibitor strand 2 at 1 μM) (Fig. 2h–k). All inhibitor is released after 24 h across conditions (Supplementary Fig. 13). When using the low yield gene 2 (nick on the non-template strand) we measured significantly slower tile activation kinetics, consistent with expectation given a much slower production of RNA activator. At high concentrations of gene 2 (100 nM) (Fig. 2l, m) and T7 RNAP (4 U/μL) (Fig. 2n, o), the tile half-activation time roughly doubles ($t_{1/2} = 120 \pm 20$ min) when compared to the high yield design. At low concentration of gene 2 we observe a consistently slower behavior of the low-yield gene, showing that nick placement allows to broaden the achievable range of transcription speeds.

To summarize, we achieved fine control over tile activation kinetics by changing the transcription speed, and we engineered switchable genetic elements to further modulate activation.

### The outputs of a genetic cascade activate distinct tiles at specific times

Natural and artificial gene networks can autonomously generate a variety of dynamic behaviors. A gene cascade, in which one gene regulates the next in a chain, is a simple architecture to generate a

temporal sequence of events synchronized with the activation of each gene. Here we build a simple cascade of two synthetic genes to sequentially control the activation of two DNA tile types. Building on the experiments we just described, we used a DNA "connector" complex to interconnect the genes 1 and 2 (Fig. 3a), which control tiles 1 and 2 respectively. Gene 1 is constitutively active, and it produces the RNA activator 1. This RNA activator plays two important roles: it activates DNA tile 1, and it simultaneously interacts with the connector complex to release a DNA molecule that activates the downstream gene 2 (Fig. 3a, Supplementary Fig. 14). An advantage of this approach is that we can control the time it takes to activate tile 2 by modularly tuning various parameters of the genetic cascade.

First, we adapted the connector design to tune the activation speed and the activation onset delay (Δt) of tile 2. We evaluated connector complexes that have the same 14-nt invading domain but differ in the length (3–6 nt) of the toehold that initiates the release of the gene DNA activator (Fig. 3a). We used conditions consistent with previous experiments (250 nM each tile 1 and 2, 1 μM RNA inhibitors 1 and 2, and 100 nM high yield synthetic genes 1 and 2). To easily measure the kinetics of tile activation, we labeled the RNA inhibitors with a fluorophore (Cy3 and Cy5 for inhibitors 1 and 2, respectively) and the tiles with their corresponding quenchers (BHQ1 and BHQ2). The speed of tile activation and the delay Δt strongly depend on the toehold length (Fig. 3b, c and Supplementary Fig. 15). Indeed, the half-

activation time ($t_{1/2}$) for tile 2 increases from 95 to 215 min when the length of the toehold domain is reduced from 6 to 3 nt (Fig. 3d, e and Supplementary Fig. 15). We verified that the DNA activator for gene 2 is displaced much faster from the connector complex by RNA output 1 when compared to the displacement of inhibitor from tile 2 (tile activation requires the additional transcription of RNA 2) (Supplementary Figs. 16 and 17). We expanded our kinetic model of tile activation and activator production to capture this interconnected system (Supplementary Note 2), and were able to reproduce the measured $t_{1/2}$ and $\Delta t$ as the connector strand displacement speed changes with the toehold length (Fig. 3d, e dashed lines). For the next experiments we designed connectors with the fastest strand displacement, i.e., 6 nt toehold. In control fluorimetry experiments in which genes were not complemented with their corresponding DNA inputs, we observed negligible activation of tiles (Supplementary Fig. 18). Thus, transcription of RNA may occur but RNA activator concentration never exceeds a minimum threshold required to activate the tiles. Similarly, we did not observe any spurious separation of DNA connector complexes with ssDNA 3′ toeholds under transcription conditions (Supplementary Fig. 19).

Next, we modulated the RNA transcription speed of gene 2 to influence the activation kinetics of tile 2, taking advantage of the high and low yield gene designs characterized earlier (Fig. 3g). Like before, we used fluorescently labeled RNA inhibitors 1 and 2 to measure their release from tiles, and we monitored their level over time under variable T7 RNAP concentration (from 1 to 4 U/μL). When using high yield genes, the release of RNA inhibitors (1, green and 2, red) is complete after about 180 min at high T7 RNAP level (4 U/μL), while the release is not completed at low T7 RNAP concentration (1 U/μL) even after 300 min (Fig. 3h). Furthermore, by using different T7 RNAP concentrations we can also control the delay $\Delta t$ in the onset of RNA production (Fig. 3i, j). When changing the T7 RNAP concentration (1–4 U/μL) we observed $\Delta t$ values ranging from 120 to 50 min (Fig. 3j, Supplementary Fig. 20). When using low yield genes, we observe a more significant $\Delta t$ of 240 min at low T7 RNAP concentration (1 U/μL), and even after 600 min we do not achieve a complete release of RNA inhibitors (Fig. 3k). As before, using a higher T7 RNAP concentration improves the release speed and reduces the delay $\Delta t$ (Fig. 3k–n, Supplementary Fig. 21). We finally tested tile activation speed when the concentration of one of the genes is kept constant, and that of the other gene varies. As expected, when the concentration of gene 2 (downstream gene in the cascade) is held constant (100 nM), as we increase the concentration of gene 1 (from 3 to 100 nM) we obtain a faster release of the RNA inhibitors while the delay $\Delta t$ decreases (Supplementary Figs. 22–24). Naturally, when the concentration of gene 1 is constant, the release of RNA inhibitor 1 remains unchanged and does not depend on the concentration of the downstream gene 2; in contrast, the release of RNA inhibitor 2 depends on the concentration of gene 2, with more inhibitor being released at higher concentrations (Supplementary Figs. 25–30). Also in this case, the measured $t_{1/2}$ and $\Delta t$ are reproduced by an unfitted kinetic model in which the parameters for tile activation and transcription are consistent across all simulations[7,40]; the low yield strategy is captured by assuming slower binding and faster unbinding of RNAP to the gene, when compared to the high yield case. Discrepancies between the model and the data can be attributed to the lack of systematic fitting, as well as to the uncertainty and batch-to-batch variability in the concentrations, activity, and half-life of enzymes. The fact that the model consistently underpredicts delays may also be due to unmodeled processes such as the accumulation of elongated or abortive RNA that may slow down assembly by creating misfolded tile complexes[31]. Nevertheless, the trends of simulations and experimental data are qualitatively consistent, confirming that the kinetics of the components involved are generally predictable and modular.

## Timing assembly and polymer organization through different genetic cascades

We showed how distinct tile types can be activated at different times by modulating the features of an autonomous two-gene cascade. Next, we examine how this fine temporal control over tile activation impacts the dynamics of tile assembly into nanotubes, as well as the properties of the nanotubes. Using the tile designs introduced earlier (1 and 2), we characterized the kinetics of nanotube growth in the presence of the two-gene cascade. In these experiments each tile carries a different molecular "load", as it is decorated with a distinct fluorophore (tile 1 with Cy3, nanotubes colorized in green; and tile 2 with Cy5, nanotubes colorized in red) which makes it possible to track assembly via fluorescence microscopy. As in prior experiments, each tile (250 nM) is initially bound to its RNA inhibitor (1 μM), and activated by their transcribed RNA activator (Supplementary Fig. 31). We first verified that nanotubes remain stable under our in vitro transcription conditions (Supplementary Fig. 32). We then verified that genes 1 (constitutively active) and 2 (switchable) individually trigger the assembly of micron-long DNA nanotubes. Figure 4a, b show that gene 1 correctly induces assembly of green nanotubes from tile 1, by producing RNA 1 that activates tile 1. These nanotubes present a mean length of 3–4 μm after about 8 h, while their density (number of nanotubes per 100 μm$^2$) decreases over time due to joining events[30] (Fig. 4c, d). It is noteworthy that the temporal evolution of the growth of the nanotubes influences the count (i.e., number of structures per 100 μm$^2$). As the average length of the nanotubes increases, their number decreases proportionally to the transcription time (Fig. 4c, d). Next, we verified that gene 2 correctly induces assembly of tile 2 into red nanotubes, in both the high and low yield variants. As expected, the high yield variant of gene 2 induces nanotube growth significantly faster than the low yield variant. Nanotubes reach their plateau length within 8 h when controlled by the high yield gene (Fig. 4f–h). It takes more than 12 h to reach a comparable mean length under the control of the low yield gene (Fig. 4i–k), which introduces a noticeable time delay before nanotubes are visible, due to the longer time it takes to build up enough RNA to activate tiles above their nucleation threshold.

We finally controlled the self-assembly of the two tile types simultaneously, under the control of our interconnected two-gene cascade (Fig. 4l). We used the constitutively active gene 1 to activate either the high yield or low yield version of gene 2 (100 nM each gene), using our connector complex (300 nM). As the cascaded genes produce their RNA activators sequentially, tiles 1 and 2 also self-assemble into nanotubes sequentially. Because tiles 1 and 2 have identical sticky ends and can interact, by tuning the speed of the cascade we can control the level of free tiles that are active at a particular point in time. If tiles 1 and 2 are activated nearly simultaneously, nanotubes should assemble from a mix of tiles yielding random green-red polymers. If tile 2 is activated much later than tile 1, green nanotubes should form first depleting free tile 1, making it possible for tile 2 to assemble into red nanotubes: in this case, we should obtain block green or red polymers. We built a fast cascade using the high yield version of gene 2: this results in rapid activation of tiles 2, which combine with unpolymerized tile 1 and yield random polymers that appear yellow in the merged channels (Fig. 4m–o). A slow cascade including the low yield version of gene 2 results in delayed and slower activation of tile 2, which exceeds the threshold for nucleation when tiles of type 1 are completely polymerized into green nanotubes. Thus, tile 2 produces red nanotubes which begin to be visible around 12 h (Fig. 4p–r). We characterized quantitatively the nanotube composition under the control of the fast and slow cascade by tracking the statistical properties of epifluorescence microscopy images in terms of the spatial localization of individual tiles within nanotubes. We calculated the Pearson coefficient (PC), which measures the strength of the linear relationship between the fluorescence intensity values of the green

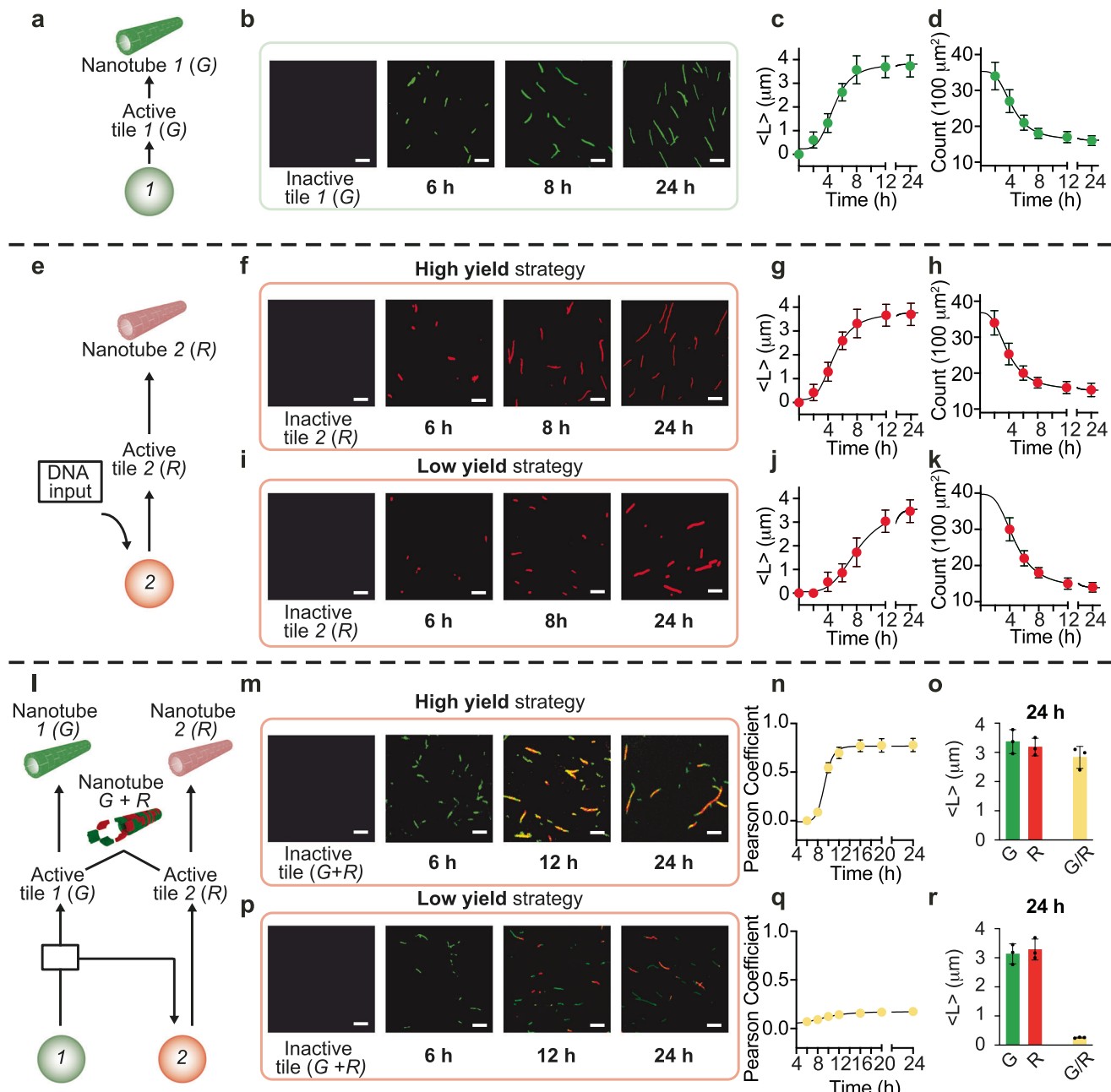

**Fig. 4 | Controlling the growth and composition of DNA nanotubes through individual and interconnected genes. a** Gene 1 (green circle) activates the corresponding tile 1 to self-assemble into green DNA nanotubes. **b** Fluorescence microscopy images of green (G) nanotubes after production of RNA activator in the presence of T7 RNAP (4 U/µl). **c** Kinetic traces of nanotube length; *<L>* indicates mean nanotube length. **d** Count (number of structures per 100 µm²) of assembled green nanotubes. **e** Gene 2 (red circle), when completed with an external DNA input, leads to the activation of tile R, which resembles red DNA nanotubes. **f** Fluorescence microscopy images of red (R) nanotubes using a high yield gene in the presence of T7 RNAP (4 U/µl). **g** Kinetic traces of nanotube length and **h** count of assembled red nanotubes. **i** Fluorescence microscopy images showing red nanotubes using a low yield gene in the presence of T7 RNAP (4 U/µl). **j** Kinetic traces of nanotube length and **k** count of assembled red nanotubes. **l** Interconnected genes 1 and 2 allow sequential activation of two different tiles (G and R) to drive the assembly of two different nanotube populations. **m** Fluorescence microscopy images of green and red random nanotubes after

production of the two RNA activators from their respective genes (gene 2 is a high yield). **n** Pearson coefficient values of reassembled random G/R nanotubes. **o** Histograms of nanotube length for each channel (green, G, red, R, and merged, G/R) after the 24 h reaction. **p** Fluorescence microscopy images of green (G) and red (R) nanotubes using gene 2 is with low yield design. **q** Pearson's coefficient values of reassembled G/R nanotubes. **r** Histograms of nanotube length for each channel (green, G, red, R, and merged, G/R) after the 24 h reaction. Experiments shown were performed using [tile G] = [tile R] = 250 nM; [RNA inhibitors] = 1 µM; [gene 1] = [gene 2] = 100 nM; [connector complex] = 300 nM. Nanotubes labeled with Cy3 (G) and Cy5 (R) dyes were imaged at a concentration of 50 nM. Black curves are logistic functions fitted to the data points as a guide to the eye (Supplementary Note 1). Scale bars for all microscope images, 2.5 µm. Experimental values represent averages of three separate measurements; error bars represent standard deviation and the center of the error bars represents the mean.

and red regions on the nanotubes. PC values close to 0 indicate low colocalization of the fluorophores on the structures, while PC values around 1 indicate high colocalization. With the fast gene cascade yielding randomly distributed nanotubes, we measured a PC = 0.80 ± 0.16, which confirms colocalization (Fig. 4n, o). For the slow gene cascade, we obtained a PC of 0.20 ± 0.02, which indicates very limited colocalization of the tiles (Fig. 4q, r).

By combining fast and slow genetic components in a cascade we can thus program the timing as well as the properties of self-assembled structures, and this approach can be scaled to a larger number of genes and assembling components, as we will show in the next sections.

Building on the lessons learned from the two-gene cascade, we demonstrate a series of temporal programs to assemble diverse types of nanotubes through a three-gene cascade shown in Fig. 5a. Again, the genes are cascaded through connector complexes that bind to the RNA activator produced by the gene upstream, and release DNA activator for the gene downstream. Like before, each gene activates specifically only one type of tile through a unique activator (gene 1 activates tile 1, colored in green; gene 2 activates tile 2, colored in red, and gene 3 activates tile 3, shown in blue), but tiles share their sticky ends and can interact. For this reason, like in the two gene cascade, the timing of activation of each tile type influences the random or block polymer organization of nanotubes.

We demonstrate four different nanotube organization patterns by using different combinations of high-yield and low-yield genes that change the speed of activation of tiles in the cascade. Using high-yield variants of both genes 2 and 3, we obtained nanotubes with a random distribution of the three different tiles. This is evident from the overlaps of the green, red, and blue channels and is also confirmed by the calculated PC over time (Fig. 5b, c). The average length of the structures measured from the merged channels (G/R, G/B, and R/B) is similar to the length measured from the individual channels, with values within the standard deviation (Fig. 5d). When a low-yield version of gene 2 is used, we observe limited co-localization of tiles 1 and 2 (green/red). However when combined with a high-yield version of gene 3, the activation of tile 3 is nearly as fast as that of tile 2, and we obtain random red/blue nanotubes (Fig. 5e–g). As expected, the combination of high-yield gene 2 and of low-yield gene 3 results in random green and red nanotubes (tiles 1 and 2) but blue block polymers (tile 3) (Fig. 5h–i). Finally, when using low-yield variants of both genes 2 and 3, only block polymers are formed since each tile type is activated slowly and is completely polymerized before the next tile in the chain becomes active (Fig. 5k–m). These experiments collectively illustrate how we can autonomously route the assembly of a material, here exemplified by DNA tiles loaded with different fluorogenic molecules, by taking advantage of interconnected gene networks and their programmable temporal responses. This approach could be scaled up to include more complex gene networks that could deliver a variety of instructions, beyond activation of the self-assembling monomers, as we show in the next section.

### Programming temporal waves of assembly of distinct nanotube populations

Temporal sequences of assembly and disassembly of biomolecular scaffolds enables many biological functions like cell growth, motility, and development. Here we take advantage of our suite of synthetic genes to control not only the appearance, but also the dissolution of multiple types of DNA nanotubes in an autonomous manner. We focus on demonstrating the transient, sequential assembly of three types of tiles each carrying a different fluorescent cargo. We engineered six cascaded genes, three of which produce RNA activators that enable tile assembly into nanotubes (genes 1–3), and the other three (genes 4–6) produce RNA inhibitors (Fig. 6a). We programmed our cascade to produce RNA inhibitors with a significant delay when compared to the activation steps. This is due to the fact that the disassembly of nanotubes occurs much faster than their assembly upon tile activation[7]. We

obtained a sufficient delay in the inhibition genes by using high yield genes at low concentration, and by slowing down the response of the connector complexes. In a control experiment we verified that a reduced toehold length in the connector complex leads to an increase in the time delay for tile disassembly, with shorter toeholds leading to longer delays (Supplementary Figs. 32–36). This was verified by testing connector complexes with toeholds between 3 and 6 nt to release DNA activator for a set of new genes (numbered 4, 5, and 6) designed to produce RNA inhibitor for tiles 1, 2, and 3. We selected a toehold length of 3 nt for all connectors activating genes 4, 5, and 6 that produce RNA inhibitors (Supplementary Figs. 37 and38). A comparable temporal response could have been achieved by combining low yield inhibitor genes and 6 nt toehold connectors.

Our six-gene cascade achieves sequential steps of assembly and disassembly of three distinct types of tiles. This autonomous succession of events was achieved by combining in one pot tiles 1, 2, and 3 (250 nM each) in a solution along with their corresponding RNA inhibitors (1 µM each), the complete set of connectors (300 nM each), and high yield gene variants (100 nM each). After addition of T7 RNAP, we observe rapid growth of green nanotubes resulting from the assembly of tile 1 (Fig. 6b, c). As gene 2 is activated, red nanotubes and random green/red nanotubes appear (Fig. 6d). Because the reactions involving connector complexes and RNA transcription proceed faster than self-assembly, we begin to observe disassembly of green nanotubes as gene 4 becomes active, before we can observe blue nanotubes (activated by gene 3) form (Fig. 6e, f). Blue nanotubes appear about 10 h after the start of the reaction, and random red/blue polymers form (Fig. 6g). As gene 5 becomes active and produces inhibitor for tile 2, the red nanotubes begin to disassemble. Next, gene 6 becomes active and induces disassembly of blue nanotubes. After 16 h the process is complete and all nanotubes have been dissolved (Fig. 6i). Additional delay in the disassembly steps may have been achieved by using low yield inhibitor genes (Fig. 3k).

## Discussion

We have demonstrated a platform of modular components to build DNA-based materials whose properties evolve according to tunable temporal programs. Taking inspiration from how cells regulate sequential developmental events[41], we have used artificial genetic cascades to produce RNA outputs that sequentially control activation or deactivation of distinct self-assembling DNA tiles. Further, our approach demonstrated that it is possible to control not only when a particular type of DNA polymer forms or dissolves, but also the polymer composition, which depends on the level of active tiles at a particular point in time. To interconnect genes and control RNA levels over time we used a strategy relying on promoter "nicking" and promoter strand displacement[20,37], however we expect that other approaches may be effective in generating nucleic acid regulators[16,42]. RNA levels could be tuned more finely by introducing RNA-degrading enzymes[43]. Our strategy takes advantage of simple DNA gates to route a single RNA output to control two processes simultaneously: the activation of DNA tiles, and the activation of downstream genes. We expect this approach could be expanded so that a single gene could control multiple processes in parallel, while maintaining modularity and minimizing the need to design additional genes.

In previous work we showed that reversible assembly of nanotubes can be obtained using dynamic circuits based on negative feedback, such as molecular oscillators and pulse generators[7,31]. However, we found both classes of circuits to be sensitive to perturbations such as competition for a limited enzyme pool and accumulation of waste species resulting from RNA transcription and degradation byproducts[40]. This sensitivity to perturbations made it difficult to achieve repeated cycles of material assembly and disassembly[40]. Undesired coupling between the nanostructures and the circuits, introduced by DNA binding enzymes, further compromises their dynamics and the modularity of the

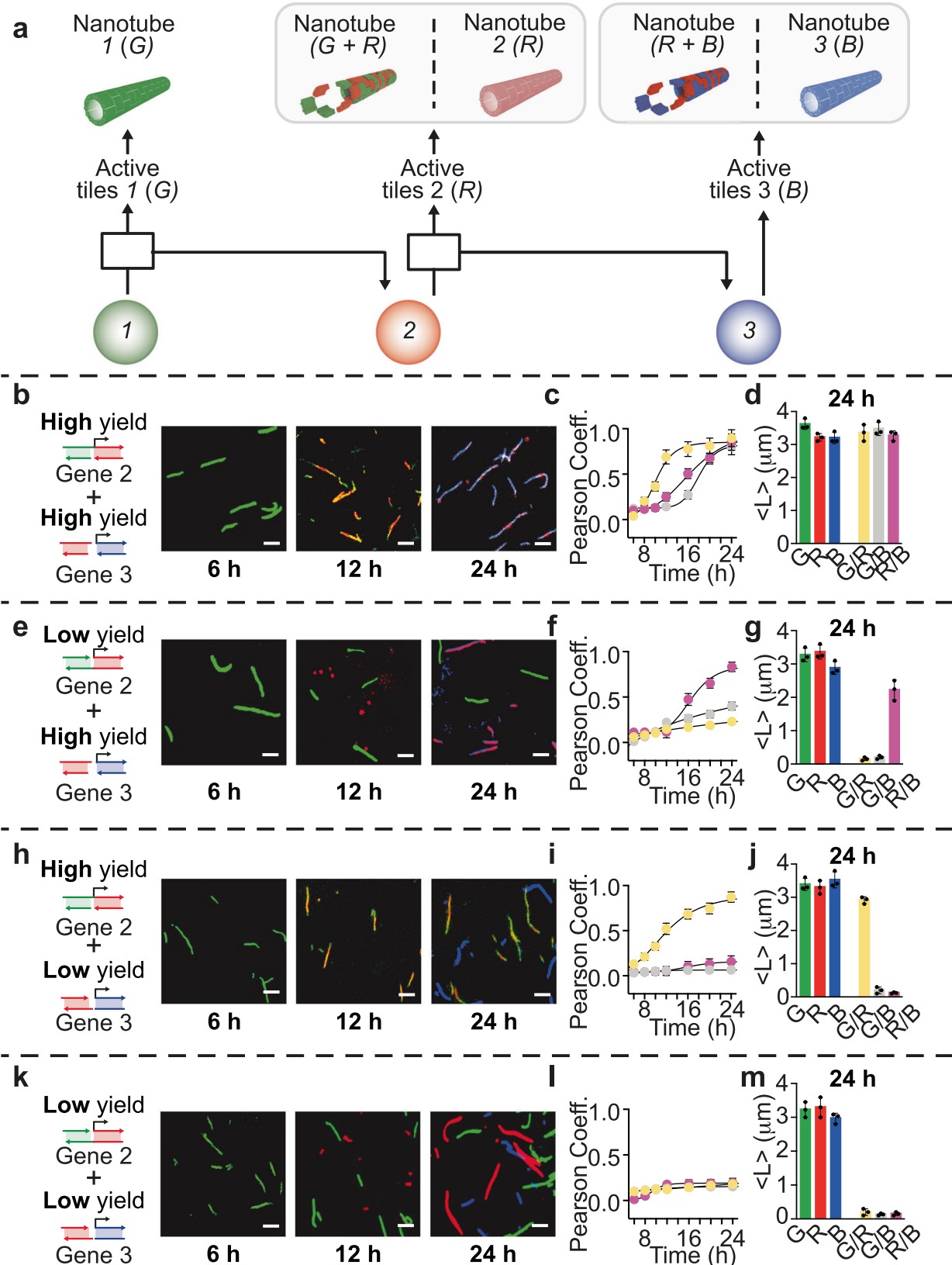

system[7]. In contrast, we found cascaded gene networks to be simple to tune and modular, and we could interconnect robustly six distinct genes to induce up to three full assembly cycles. Our thorough characterization of the temporal response of each circuit component is useful toward further expanding the achievable dynamics. The duration and peak of each assembly cycle could be systematically tuned by changing the order of genes in the cascade, the concentration and yield of each gene, as well as the concentration and toehold length of connectors. For example, the combination of low yield genes and short-toehold connectors could maximize the delay between nanotube assembly events illustrated in Fig. 6, creating well separated waves of nanotube formation and expanding the temporal range of events.

**Fig. 5 | Development of different nanotube populations by tuning the production kinetics of distinct RNA activators. a** Three interconnected genes are activated by their respective upstream nodes at a specific time point, resulting in the formation of distinct nanotube populations. **b** Fluorescence microscopy images of green (G), red (R), and blue (B) nanotubes after production of the corresponding RNA activators using a combination of both high yield genes 2 and 3. **c** Pearson coefficient of reassembled G/R/B nanotubes at different reaction times. **d** Histograms of nanotube length for each channel (green, G, red, R, blue, B and merged, G/R, G/B and R/B) after the 24 h reaction. **e** Fluorescence microscopy images of green, red, and blue nanotubes using a combination of low yield gene 2 and high yield gene 3. **f** Pearson coefficient of reassembled G/R/B nanotubes at different times. **g** Histograms of nanotube length for each channel after the 24 h reaction. **h** Fluorescence microscopy images of green, red, and blue nanotubes using a combination of high yield gene 2 and low yield gene 3. **i** Pearson coefficient of reassembled G/R/B nanotubes at different times. **j** Histograms of nanotube length for each channel after the 24 h reaction. **k** Fluorescence microscopy images of green, red, and blue nanotubes using a combination of both low yield genes 2 and 3. **l** Pearson coefficient of reassembled G/R/B nanotubes at different times. **m** Histograms of nanotube length for each channel after the 24 h reaction. Experiments shown were performed using [tile G] = [tile R] = [tile B] = 250 nM; [RNA inhibitors] = 1 μM; [gene 1] = [gene 2] = [gene 3] 100 nM; [connector complexes] = 300 nM. Experimental values were calculated via triplicate experiments, and error bars reflect standard deviations. Scale bars for all microscope images, 2.5 μm. In (**b**, **e**, **h**, **k**), nanotubes (G), (R), and (B) were imaged at a concentration of 50 nM. In (**c**, **f**, **i**, **l**), the G/R channel gives a yellow color, G/B gives a gray color, and R/B gives a pink color. Experimental values represent averages of three separate measurements; error bars represent standard deviation and the center of the error bars represents the mean.

Our demonstration takes advantage of the versatility and composability of nucleic acid components, which have made it possible to build rapidly growing libraries of circuits[44] and structures[19], and of systems that integrate them[7,8,45]. While here we focused on using genetic cascades to regulate the assembly of nanotubes as a simple, model DNA nanostructure, we expect that these cascades could be repurposed to coordinate the assembly of very different structural parts, like DNA origami[46,47], nanoparticles[48,49], and coacervates[50]. Overall, our work suggests a way toward scaling up the complexity of biomolecular materials by taking advantage of the timing of "molecular instructions" for self-assembly, rather than by increasing the number of molecules carrying such instructions. This points to the exciting possibility of generating distinct materials that can spontaneously "develop" from the same finite set of parts, by simply rewiring the elements that control the temporal order of assembly.

## Methods
### Buffers, oligonucleotides, and enzymes
Reagent-grade chemicals (Trizma base (cat. #93352), acetic acid (cat. #A6283), boric acid (cat. #B0252), Ethylenediaminetetraacetic acid (EDTA) (cat. #798681), acrylamide-bis-acrylamide (40%) (cat. #A7168), ammonium persulfate (APS) (cat. #A7460), and N,N,N[I],N[I]-tetramethyl ethylenediamine (TEMED) (cat. #T9281)) were purchased from Sigma-Aldrich (St Louis, Missouri) and used without further purification. The ORangeRuler 50 bp DNA ladder was purchased from ThermoFisher Scientific (USA) (cat. #SM0613). Transcription buffer was purchased from ThermoFisher Scientific (USA) and stored at −20 °C. 5× transcription buffer contains: 200 mM Tris–HCl, 30 mM MgCl$_2$, 50 mM DTT, 50 mM NaCl, and 10 mM spermidine (pH 8.0 at 25 °C). Nucleotide triphosphates (NTPs) were purchased from Jena Bioscience (Germany) and stored at −20 °C.

HPLC-purified oligonucleotides were purchased from IDT DNA Technologies (Coralville, IA) and Metabion International AG (Planegg, Germany). All oligonucleotides were dissolved in double-distilled H$_2$O at a concentration of 100 μM and aliquoted at −20 °C for long-term storage. All sequences were designed using Nupack or IDT software programs and are reported in the Supplementary Tables[1,2]. The concentration of oligonucleotides was quantified by UV–Vis spectrophotometry using a Thermo Fisher Scientific Multiskan SkyHigh Microplate Spectrophotometer.

T7 RNA polymerase was purchased from ThermoFisher Scientific (USA) (cat. #EP0113). The bulk batch of T7 RNA polymerase was split into smaller aliquots (enough for 20–30 experiments) to minimize degradation of the enzyme by repeated removal from the freezer, and stored at −20 °C.

### Fluorescence measurements
All fluorescence experiments were performed at 30 °C, in 1× transcription buffer, 10 mM NTPs, pH 8.0 in a 100 μL cuvette (total volume of solution 100 μL). Equilibrium fluorescence measurements were performed using a Cary Eclipse fluorimeter. Excitation was at 550 (±5) nm and acquisition at 570 (±5) nm (for strands labeled with Cy3) or at 650 (±5) nm and acquisition at 670 (±5) nm (for strands labeled with Cy5).

### Fluorescence measurements for tile activation
The self-assembly of DNA tiles driven by RNA activator transcription was studied by first examining the assembly of individual tiles (1 and 2) at a fixed concentration of RNA inhibitor strand (1, and 2, 1 μM) in the presence of different concentrations of the transcribing gene for the RNA activator strand and the T7 RNA polymerase for both strategies described in the main text. Similarly, to study the assembly of DNA nanotubes driven by green and red activators, equimolar concentrations of inactive tiles 1 and 2 (250 nM labeled tiles/1 μM RNA inhibitor) were mixed in the absence or presence of the connector system (300 nM) in the same buffer solution at different concentrations of the transcribing genes for the RNA activator strands and the T7 RNA polymerase enzyme. The percentage of tile activation, and thus inhibitor release, was tracked with inactive tiles labeled with a fluorophore/quencher pair so that tile activation could be tracked by the increase in fluorescence signal due to inhibitor release. We recorded the fluorescence signal in real time until it reached equilibrium before the addition of the T7 RNA polymerase enzyme.

### Data analysis for tile activation
The values for RNA inhibitor released from tiles (nM) are calculated from the relative fluorescence signal registered upon addition of a saturating concentration of T7 RNA polymerase (4 U/μL) in the presence of the specific activator gene according to the following formula:

$$[Released\ Inhibitor] = \frac{[Inhibitor] * (F_{+T7RNAP} - F_{-T7RNAP})}{F_{max}}$$

where $F_{+T7\ RNAP}$ is the fluorescence signal observed after the addition of T7 RNA polymerase in the presence of a fixed concentration of activator gene; $F_{-T7\ RNAP}$ is the fluorescence signal observed in the absence of T7 RNA polymerase at a fixed concentration of activator gene; $F_{max}$ is the maximum fluorescence value that can be achieved when 250 nM of inhibitor is released from the DNA tiles; [Inhibitor] is 250 nM and is the maximum concentration that can be released from the DNA tiles (250 nM).

### Active tile preparation
DNA tiles for all systems presented were prepared as follows. The solution of active tiles was prepared to a target concentration of 5 μM by mixing S1–S5 strands of each tile type (1, 2, and 3) in stoichiometric ratios with 1× transcription buffer and double distilled water (ddH$_2$O) in DNA Lo-bind tubes. The solution was annealed using an Eppendorf Mastercycler Gradient Thermocycler by heating to 90 °C and cooling to 25 °C at a constant rate for a period of 6 h. The annealed tiles were

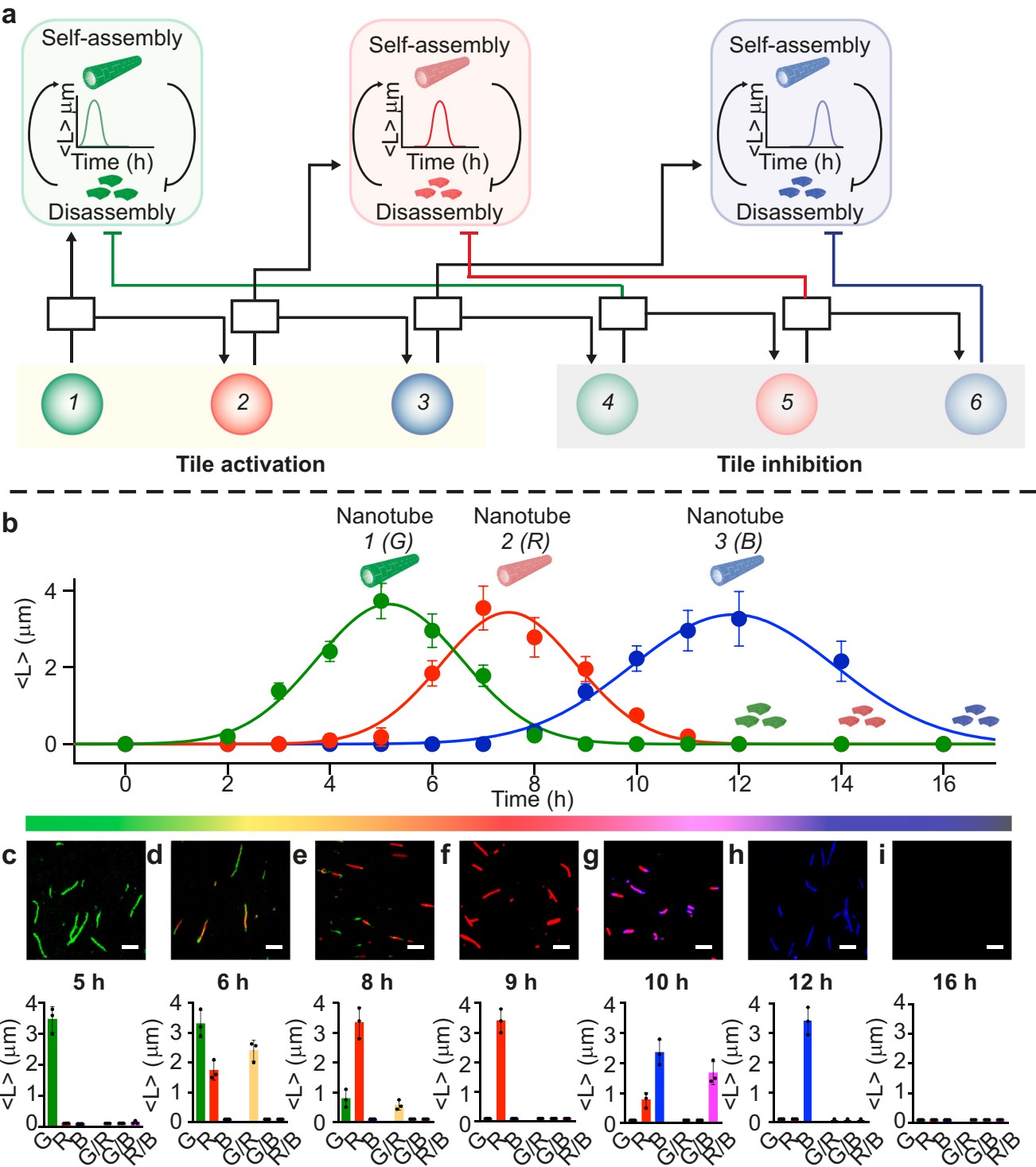

**Fig. 6 | Autonomous, transient temporal waves of distinct nanotube populations driven by a genetic cascade. a** We designed six interconnected genes (filled colored circles) that can control the self-assembly (the first three) or disassembly (the second three) of DNA nanotubes by transcribing specific RNA activators or inhibitors. **b** When the specific RNA activator displaces the inhibitor from the corresponding tile, the activation process becomes dominant and promotes nanotube regrowth. In contrast, the production of RNA inhibitors leads to their gradual degradation. Kinetic traces show the autonomous assembly and disassembly of G, R, and B nanotubes (mean length, <*L*>) at different reaction time points. **c–i** Fluorescence microscopy images show that the mean nanotube length increases when growth is triggered by transcription of RNA activators and decreases when transcription of inhibitors begins. Histograms of nanotube length

(<*L*>/μM) measured from fluorescence microscopy images for each channel (green, G, red, R, blue, B and merged, G/R, G/B, and R/B). Experiments shown were performed using [tile G] = [tile R] = [tile B] = 250 nM; [RNA inhibitors] = 1 μM; [gene 1] = [gene 2] = [gene 3] = 100 nM; [gene 4] = [gene 5] = [gene 6] = 3 nM; [connector complexes] = 300 nM; Scale bars for all microscope images, 2.5 μm. Nanotubes (G), (R), and (B) were imaged at a concentration of 50 nM. In (**b–i**) the mean and s.d. of nanotube length are calculated over triplicate experiments. The G/R channel gives a yellow color, G/B a gray color, and R/B a pink color. In (**b**), green, red, and blue lines are Gaussian functions fitted to data points as a guide to the eye. Experimental values represent averages of three separate measurements; error bars represent standard deviation and the center of the error bars represents the mean.

then diluted to a final concentration of 250 nM in 1× transcription buffer. The solution of the assembled tiles was diluted to 50 nM in the same buffer before a microscope image was taken.

### Inactive tile preparation

The annealed tiles (1, 2, and 3) were diluted in 1× transcription buffer to a final concentration of 250 nM each, and incubated in the Mastercycler at 30 °C for 1 h prior to adding the RNA inhibitor strand. The assembled tiles solution was diluted to 50 nM in the same buffer before a microscope image was taken. The inhibitor strands for tiles 1, 2, and 3 (G, R, and B) were then added to an excess concentration of 1 μM to ensure the complete disassembly of all DNA tiles. The reaction was allowed to proceed for 30 min at 30 °C and then we imaged the samples using a fluorescence microscope.

### Reassembly by addition of synthetic RNA activator strand

The annealed tiles were diluted to a final concentration of each tile of 250 nM. The RNA inhibitor 1 (G) was then added at a concentration of 1 μM and the microscope image was taken after 30 min. The RNA inhibitor 2 (R) was added to a separate solution at a concentration of 1 μM and the microscopy image was taken after 30 min. The RNA inhibitor 3 (B) was added to a separate solution at a concentration of 1 μM and the microscope image was taken after 30 min. The RNA activators 1, 2 and 3 (G, R, and B) were then respectively added at a concentration of 3 μM and the microscope image was taken for the next 24 h. The same protocol was followed for mixed G/R/B nanotubes reassembly. The RNA inhibitors 1, 2 and 3 (G, R, and B) were added at a concentration of 1 μM to the annealed tiles (G, R, and B), respectively. Samples were incubated at 30 °C and a confocal image was taken after 1 h. Of note, the tiles disassemble. The RNA activators 1, 2, and 3 (G, R, and B) were then added at a concentration of 3 μM and the confocal images were taken in the next 24 h.

### Reassembly via transcription of RNA activator strand

Inactive tiles 1, 2, and 3 (G, R, and B, respectively) were incubated at 30 °C for 1 h before the addition of T7 RNA polymerase to produce the RNA activator strand. The annealed synthetic gene for transcribing the corresponding RNA activator strand (1, 2, and 3) was mixed with the inactive tiles (250 nM each) and transcription mix (T7 RNAP 4 U/μL, 1× transcription buffer, 10 mM each NTPs) at 30 °C and observed under a fluorescence microscope for several hours.

### Disassembly and assembly via transcription of RNA regulators

Inactive tiles G, R, and B were incubated at 30 °C for 1 h prior to production of the RNA activator strand. The annealed synthetic gene for transcribing the corresponding RNA activator (G, R, and B) and inhibitor strands were mixed with the inactive tiles (250 nM each), connector complexes (300 nM), and transcription mix (T7 RNAP, 1× transcription buffer, 10 mM each NTPs) at 30 °C and observed under a fluorescence microscope for several hours.

### Fluorescence microscopy experiments

Fluorescence imaging of DNA nanostructures for fluorescence microscopy, the central strand of each tile (S3) was labeled at the 5′ end with a different fluorophore (Cy3, Cy5, or 6-FAM). Nanotube samples were imaged using an inverted microscope (Zeiss Observer 7) with a 100× oil immersion objective (EC Plan-Neo Fluor) and a monochrome Axiocam 305 camera. For imaging, samples were diluted with the experimental buffer to a final concentration of 50 nM of each tile. A 5 μL drop of this diluted solution was then deposited between clean microscope coverslips (Menzel–Glaser; thickness: 0.13–0.16 mm; size: 18 × 18 mm). Images were acquired using 90 HE LED filters (EX: 385, 475, 555, 630 nm; QBS 405 + 493 + 575 + 653; EM: QBP 425/30 + 514/30 + 592/30 + 709/100). The exposure time was set to 10000 ms. Ten images at the same location were processed to correct for uneven illumination and superimposed to produce multicolor images using ZEN 3.6 software (ZEISS).

### Fluorescence microscopy data and image processing

Images were analyzed and processed using ZEN 3.6 software (ZEISS) to correct for uneven illumination and overlaid to produce multicolor images. Branched or looping nanotubes were removed from the length dataset using an in-house MATLAB script. ImageJ was used to analyze the pixel intensity of the selected structure for each individual channel. Colocalization analysis was performed using a plugin for ImageJ software called JACoP (Just Another Co-localization Plugin)[51,52].

### Native PAGE experiments

Native polyacrylamide gels (15%) were prepared with TBE (10×), APS, and TEMED at appropriate ratios. Gels were cast in 10 × 10 cm, 1.5 mm thick disposable minigel cassettes and allowed to polymerize for at least 30 min before electrophoresis. The gel was incubated with a running buffer (1× TBE solution, pH 8.0) for 30 min at 25 °C. Sample volumes of 10 μl were combined with 1 μl of 6× Orange DNA Loading Dye and then loaded directly into the gel. Native PAGE was run in a mini-PROTEAN tetracell electrophoresis unit (Bio-Rad) at 25 °C using 1× TBE buffer at pH 8.0 and a constant voltage of 110 V for 2 h 30 min (using the Bio-Rad PowerPac Basic power supply). Gels were stained with SYBR Gold and scanned using a ChemiDoc MP imaging system (Bio-Rad).

### Statistics and reproducibility

All data shown in graphs are presented as mean ± standard deviation. Statistical analysis was performed using GraphPad Prism 8 software. No specific preprocessing of data was performed prior to statistical analyses. Statistics on nanotube length and number were obtained from fluorescence microscopy experiments. To gather these statistics, we collected 10 separate fields of view at each time point, for each of three experimental replicates.

### Reporting summary

Further information on research design is available in the Nature Portfolio Reporting Summary linked to this article.

## Data availability

Source data are provided with this paper.

## Code availability

The MATLAB code developed in-house for the measurement of nanotube length and the MATLAB code for reproducing simulations in Figs. 2 and 3 is available in the GitHub repository (https://github.com/FrancoLabUCLA/Sorrentino-2024-Nat-Commun). A DOI for the GitHub repository can be found through zenodo (https://doi.org/10.5281/zenodo.13710710).

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

## Acknowledgements

Experimental work at UCLA was supported by the US Department of Energy (DOE), Office of Science, Basic Energy Sciences (BES) under

Award # DE-SC0010595 to EF. Computational modeling work was supported by the National Science Foundation (NSF) under CCF award number # 2107483 to E.F. This work was supported by the European Research Council, ERC (project n.819160) (F.R.), by Associazione Italiana per la Ricerca sul Cancro, AIRC (project n. 21965) (F.R.), by the Italian Ministry of University and Research (Project of National Interest, PRIN, P2022ANCEK; Project Fare 2020, R20FMF2B3P) and by "PNRR M4C2-Investimento 1.4- CN00000041" financed by NextGenerationEU (F.R and S.R.). D.S. is supported by a postdoctoral fellowship from Associazione Italiana per la Ricerca sul Cancro, AIRC.

## Author contributions

D.S., FR and E.F. designed research. D.S. performed experiments and analyzed data. D.S and S.R. analyzed data. E.F. designed and conducted computational simulations. D.S. developed image processing methods and analyzed data. All authors contributed to writing the paper.

## Competing interests

The authors declare no competing interests.
