## [Peer Review File · Nature Communications]

REVIEWERS' COMMENTS

Reviewer #1 (Remarks to the Author):

The revised manuscript has sufficiently addressed my questions and concerns. The additional experimental details and schematics added to the SI greatly help with clarity. I also appreciate the authors including additional controls to show minimal non-specific reactions with T7 RNAP. Regarding the absence of non-specific interactions in this paper compared to other studies, I noticed that all reactions in this manuscript were conducted at 30 C rather than 37 C. This might be a factor in reducing non-specific transcription and this might be worth noting. I support publication of this revised manuscript in Nature Communications.

Reviewer #1 (Remarks on code availability):

The authors provide logistic, exponential, and Gaussian fits to data to aid with data visualization in their figures, but the code for generating these fits doesn't appear to be provided on GitHub so exactly reproducing these fits could be challenging.

Reviewer #2 (Remarks to the Author):

I recommend the publication of the manuscript.
The authors have addressed all the issues and comments I raised in the previous review round and their revisions have significantly improved the manuscript.

Reviewer #3 (Remarks to the Author):

The authors have appropriately addressed all of this reviewer's comments.
It is recommended to accept the manuscript for publication in Nature Communications.